# COVID-19 Vaccines for Adults and Children with Autoimmune Gut or Liver Disease

**DOI:** 10.3390/vaccines10122075

**Published:** 2022-12-05

**Authors:** Monika Peshevska-Sekulovska, Plamena Bakalova, Violeta Snegarova, Snezhina Lazova, Tsvetelina Velikova

**Affiliations:** 1Department of Gastroenterology, University Hospital Lozenetz, 1407 Sofia, Bulgaria; 2Medical Faculty, Sofia University St. Kliment Ohridski, 1407 Sofia, Bulgaria; 3Clinic of Internal Diseases, Naval Hospital—Varna, Military Medical Academy, Medical Faculty, Medical University, 9000 Varna, Bulgaria; 4Pediatric Department, University Hospital “N. I. Pirogov”,“General Eduard I. Totleben” Blvd 21, Health Care Department, 1606 Sofia, Bulgaria; 5Faculty of Public Health, Medical University Sofia, Bialo More 8 Str., 1527 Sofia, Bulgaria; 6Department of Clinical Immunology, University Hospital Lozenetz, Medical Faculty, Sofia University St. Kliment Ohridski, 1407 Sofia, Bulgaria

**Keywords:** autoimmune gut disease, autoimmune liver disease, COVID-19 vaccine, COVID-19, adults, children, recommendations

## Abstract

The SARS-CoV-2 pandemic raised many challenges for all patients with chronic conditions and those with autoimmune diseases, both adults and children. Special attention is paid to their immunological status, concomitant diseases, and the need for immunosuppressive therapy. All of these factors may impact their COVID-19 course and outcome. COVID-19 vaccination is accepted as one of the most successful strategies for pandemic control. However, individuals with immune-mediated chronic diseases, including autoimmune liver and gut diseases, have been excluded from the vaccine clinical trials. Therefore, we rely on real-world data from vaccination after vaccine approval for these patients to fill the evidence gap for the long-term safety and efficacy of COVID-19 vaccines in patients with autoimmune gut and liver diseases. Current recommendations from inflammatory bowel disease (IBD) societies suggest COVID-19 vaccination in children older than 5 years old, adults and even pregnant females with IBD. The same recommendations are applied to patients with autoimmune liver diseases. Nevertheless, autoimmune disease patients still experience high levels of COVID-19 vaccine hesitancy, and more studies have to be conducted to clarify this issue.

## 1. Introduction

The culpable agent for the coronavirus disease 2019 (COVID-19) pandemic is the severe acute respiratory syndrome coronavirus-2 (SARS-CoV-2) [1]. It causes acute respiratory distress syndrome (ARDS) in a large number of individuals with severe pulmonary injury [2]. However, the pandemic has raised significant concerns about managing immunocompromised patients. Recent research reveals that these patients have more severe disease courses due to their underlying changed immunological status and immunosuppressive medicines [3]. Furthermore, Kim et al. reported a 40% additional risk for in-hospital mortality and 30% for intensive care unit (ICU) admission among these patients [4].

In response to the high mortality rate and casualties of the COVID-19 pandemic globally, pharmaceutical companies have produced effective vaccines against the SARS-CoV-2 infection. Therefore, immunization against SARS-CoV-2 is considered the most suitable tool in the hands of physicians. Several vaccine candidates and tactics were developed shortly after the onset of SARS-CoV-2 pandemic [5]. However, they were not fully successful in managing or controlling the COVID-19 pandemic. The spread of COVID-19 is mostly influenced by the appearance of novel viral variants brought on by the acquisition of genetic alterations in SARS-CoV-2 in various regions of the world and subsequent rapid transmission across the continents [5].

Additionally, depending on the population and variations, the efficacy of the approved vaccinations against the newly emerging SARS-CoV-2 mutations varied. This highlights the need for a broad-spectrum vaccine that could elicit a more effective immune response toward all new variants [6].

Different types of COVID-19 vaccines have been developed, including live-attenuated vaccines, protein-based vaccines, and gene vaccines (mRNA, vector-based, and VLPs) [5,6]. Nevertheless, the hastened course of vaccine development raises several concerns, especially in particular groups of patients, such as those with an altered immune system. Moreover, many individuals with immune-mediated chronic diseases, including autoimmune liver and gut diseases, have been excluded from vaccine clinical trials [7]. This resulted in an evidence gap for the long-term safety and efficacy of COVID-19 vaccines in patients with autoimmune diseases.

Some of the theoretical concerns are related to the complex pathogenesis of autoimmune diseases, including liver and gut conditions, immunosuppressive therapy, and the cumulative risk of flare [8]. We rely on real-world data from vaccination after vaccine approval for these patients.

Nevertheless, autoimmune disease patients still experience high levels of COVID-19 vaccine hesitancy. Some concerns are stopping immunosuppression before vaccination and minimizing the risk of relapse and adverse effects. The COVID-19 Vaccination in Autoimmune Diseases (COVAD) study, a long-term ongoing global self-reported study that includes patients with autoimmune disease, collected data on short and long-term adverse effects and disease flares in patients following COVID-19 vaccines [9,10].

A questionnaire-based study among Chinese people with IBD also showed COVID-19 vaccination hesitancy, mostly related to the history of immune-modifying therapies, potential adverse reactions, and effectiveness [11].

In this background, little is known about the vaccines’ safety profiles, effectiveness, adverse effects, and infection flare in adult and pediatric patients with autoimmune liver and gut diseases. Therefore, our mini-review aims to assess the efficacy and safety of SARS-CoV-2 immunization to prevent developing severe COVID-19 infection in this cohort of patients without interfering with their current immunosuppressive therapy.

## 2. Autoimmune Gut Diseases and COVID-19 Vaccines

Autoimmune diseases are characterized by hyperreactivity of the immune system and loss of immune tolerance, which damage and destroy healthy tissues, cells, and organs [12].

Patients diagnosed with inflammatory bowel disease (IBD), especially those who have undergone biological or immunosuppressive therapy, are a subject of interest due to the risk factors that arise from the illness regarding COVID-19 morbidity [13]. Currently, risk factors associated with a higher risk of infection due to IBD morbidity are nutrition status, age, comorbidity, and pharmacotherapy. The most enduring hypothesis is that malnutrition and food deficiency lead to a compromised immune response. Food deficiency relates to decreased phagocyte function and abnormal cell-mediated immunity [13]. Additionally, patients suffering from IBD often have vitamin D deficiency, increasing the severity of COVID-19 infection [14]. Going further down the road, we reviewed the role of vitamin D in deficient patients not only for COVID-19 severity but as a potential adjuvant for COVID-19 vaccination, as seen for other vaccines [15].

Many previously described studies underline that treating IBD patients with immunomodulators (TNF-antagonists, non-TNF targeted biologics), immunosuppressive therapy, or corticosteroids can increase the risk of infections, or the complications associated with various infections [16]. On the other hand, managing the activity of IBD is also of paramount importance because it plays a risk factor for infections or associated complications. That is why IBD treatment should be as optimal as possible and the treatment course uninterrupted [17,18].

Up to now, there are some clinical studies that involve patients with autoimmune gut diseases and assess the safety and effectiveness of COVID-19 vaccines. Thus, we conducted a literature review and summarize our findings in Table 1.

One of the first that reported on the COVID-19 vaccine and patients with autoimmune gut disease was Botwin et al. They discovered that only 3% had severe adverse events (that affect daily activity), mainly presented by malaise [19]. Three patients needed hospitalization (one of them for gastrointestinal complaints). Studies demonstrated that despite common adverse effects in the general population after COVID-19 vaccination, most of which are mild and local site reactions, there is a noticeable percentage, especially in patients with autoimmune rheumatic diseases [18,20,42]. The latter group mainly complained of localized pain (70.2%), fatigue (34.7%), headache (30.6%), and muscle ache (29.3%), with no serious adverse effects. Similar side effects were observed in IBD patients (Table 1).

Additionally, it was confirmed that IBD patients on advanced therapy with biological agents showed lesser adverse effects than non-IBD patients using other treatment strategies. No data exist comparing IBD patients with other patients with autoimmune diseases, i.e., rheumatoid arthritis and other rheumatic diseases. Regarding the general population, the side effects are comparable [19].

However, they also exerted decreased antibody production while on infliximab and vedolizumab [21,24]. Thus, Botwin et al.’s findings are helpful for physicians and patients by confirming the similar safety profile for mRNA vaccines for IBD patients [19]. Furthermore, the authors observed a difference in adverse reactions following the first and second doses. Higher rates of adverse reactions were found after the second dose [18,19]. However, patients who recovered from COVID-19 had more reactions than patients with no previous antispike response after the first dose but not the second one. Nevertheless, more studies are needed to confirm this observation.

Lev-Tzion et al. confirmed that the incidence rate of disease exacerbation after COVID-19 vaccination is comparable to unvaccinated IBD patients. Moreover, immunosuppressive treatment did not influence the effectiveness of the Pfizer-BioNTech BNT162 b2 vaccine [23].

Additionally, patients on immunosuppression may have reduced immune response (i.e., anti-TNFa but not azathioprine; for corticosteroids—depending on the dose) [43]. Furthermore, the Crohn’s and Colitis Foundation’s statement supports the COVID-19 vaccine in IBD patients without measuring antibody levels [44,45].

Other ongoing studies on the safety and effectiveness of COVID-19 in IBD patients are CORALE-Vaccine IBD (https://www.corale-study.org/ibd, accessed on 30 November 2022) and PREVENT COVID (with children, https://www.ibdpartners.org/preventcovid, accessed on 30 November 2022). 

Thus, the Center for Disease Control and the American College of Obstetricians and gynecologists recommend COVID-19 vaccination in female IBD patients planning pregnancy and currently pregnant or lactating women [44]. However, solid organ transplantation patients may not receive sufficient protection after vaccination.

D’Amico et al. suggested more pros than cons for SARS-CoV-2 vaccination in IBD patients. However, despite insufficient data, we can extrapolate information from data reported in patients with other autoimmune diseases [46]. This is why the American College of Rheumatology recommends COVID-19 vaccines for patients with autoimmune inflammatory diseases based on the previously available data for other vaccines [47].

The recently published systemic review and meta-analysis by Sung et al. focused on efficacy, seroconversion rate (antibody titer against SARS-CoV-2 S protein), and the adverse effects in 27,454 IBD patients from 11 studies. As expected from the data from other studies, COVID-19 incidence was comparable in IBD and non-IBD patients and 8.63 times lower than observed in unvaccinated IBD patients. However, the reported adverse event rate after vaccination was 69%, the severe adverse rate was 3%, and mortality was 0% [48].

Doherty et al. published similar results in their paper—reduced immune responses after vaccination in IBD patients on anti-TNF therapy and other immunomodulators. However, the overall conclusion is that patients with IBD still benefit from COVID-19 vaccination, and recommendations included minimizing corticosteroid doses before vaccination if possible [49].

Jena et al. in their systematic review and meta-analysis on effectiveness and durability of COVID-19 vaccination in IBD patients confirmed the lower pooled seroconversion rate. However, the pooled relative risk of infection breakthrough was similar to control subjects [50]. Tabesh et al. also conducted a systematic scoping review of 15 studies, concluding that COVID-19 vaccines are effective and safe for patients with IBD on different therapeutic regimens [51].

One of the most significant limitations of the published studies so far is that they cover effectiveness or adverse effects solely, but rarely both. Additionally, adapted immunization techniques may be appropriate in some IBD patients to maximize immunogenicity, according to prior experience [52,53].

Still, the main concerns for patients with IBD remain a lack of immune protection after vaccination (17.6% of respondents), worsened adverse effects (24.6%) due to IBD, and flare following vaccination (21.1%), according to an international web-based survey [33].

Duong et al. also reported that 2/3 of surveyed IBD patients were willing to get vaccinated against COVID-19 [54], which was also reported by Hudhud et al. [55].

## 3. Autoimmune Liver Diseases and COVID-19 Vaccines

With the current COVID-19 pandemic on the one hand and the high percentage of liver diseases worldwide on the other, the question of SARS-CoV-2 immunization in this cohort of patients arises among physicians. Consequently, they are in a rat race to assess accurate information to help their patients.

Many reports have been published concerning the severity of COVID-19, the mortality rate, and the disease outcome in patients with liver disease [56]. Regardless of the etiologic cause, all patients with chronic liver disorders are at risk of severe COVID-19 with liver failure and other complications, i.e., cytokine storm [57].

However, the use of immunosuppressants in those patients with autoimmune liver diseases appears to be strongly associated with the severity of COVID-19. Immunosuppressive drugs, such as glucocorticoids, thiopurines, mycophenolate mofetil, and tacrolimus worsen COVID-19 severity and prognosis [58].

Additionally, it was confirmed that SARS-CoV-2 can induce autoimmunity through different mechanisms, including overstimulation of the immune system, formation of excessive neutrophil extracellular traps, development of autoantibodies, etc. [59].

An interesting multicenter network study by Singh et al. enlightened the possible reasons for poor outcomes in patients with preexisting liver disease and SARS-CoV-2 infection [60]. They compared the course of the infection and the outcome in patients with and without previous liver injury. The authors point out that patients in the liver disease group had a higher risk of mortality (*p* < 0.001) due to increased cirrhosis-induced proinflammatory cytokine production and the concomitant hepatopulmonary syndrome or portopulmonary hypertension, which carry a risk of respiratory failure [60].

Cirrhosis-associated immune dysfunction (CAID) refers to patients with severe liver disease who exhibit innate and humoral immunity impairments. Although the association with severe bacterial infections has already been elucidated, CAID has also been demonstrated to predispose patients to a number of viral or fungal diseases [61]. According to Marjot et al., this aforesaid immune failure was the main culprit for some severe COVID-19 consequences found in patients with decompensated cirrhosis and is culpable for the decreased immunological responses seen with existing vaccines [62]. The authors underline the significance of CAID with an example of reduced duration of humoral immunity and HBV seroconversion rates after HBV, pneumococcal, and influenza vaccination in cirrhotic patients. Based on this evidence, patients with advanced liver disease are prone to have suppressed immunological responses to SARS-CoV-2 immunization [63].

With respect to autoimmune liver diseases (ALD), a plethora of studies have reported that patients appear to be at a higher risk of infection in general, which carries an increased mortality risk [64,65].

A similar assumption has been made regarding autoimmune-induced hepatitis (AIH) and the SARS-CoV-2 virus. However, contrary to this belief, Di Giorgio et al. found that AIH patients have the exact prevalence of COVID-19 infection as the general population [66]. There have been a few occurrences of autoimmune hepatitis (AIH) following SARS-CoV-2 vaccination, all of which were entirely cured by steroid therapy. However, more data are required to support the causation [56].

The systemic review of Alhumad et al. included 275 cases from 118 articles. It demonstrated the onset of 138 cases of autoimmune hepatitis, 52 cases of portal vein thrombosis, 26 cases of elevated liver enzymes, and 21 cases of liver injury following COVID-19 vaccination [67].

However, most patients recovered fully after treatment, without serious complications or need of long-term hepatic therapy, and the causality relationship cannot be confirmed. Moreover, the number of such cases is very small compared to the millions of vaccinations, where the protective benefits outweigh the risks.

Because patients with cirrhosis are particularly vulnerable to a severe COVID-19 course and have a high mortality rate (70% in patients with Child-Pugh C), vaccination against SARS-CoV-2 should start as soon as possible [62].

Many studies have assessed the efficacy of different types of COVID-19 vaccines and their safety profiles. However, despite the inclusion of nearly 100,000 participants in the first COVID-19 conducted trials, data for patients with liver disease are extremely limited, similar to the situation mentioned above for IBD patients and overall for patients with autoimmune, inflammatory, or immune-mediated diseases. For example, in the Pfizer vaccination study, only 0.6% of the participants had liver disease, and even fewer had moderate-to-severe liver disease (<0.1%) [68]. Approximately the same percentage of patients with liver disease were also included in the Moderna trial (0.6%) [69]. We summarized COVID-19 vaccine studies on safety profile, efficacy, and adverse effects rate in pa-tients with autoimmune liver disease in Table 2.

Moreover, the Astra-Zeneca vaccine trials explicitly omitted patients with preexisting liver pathology [71]. In addition, all trials excluded the patients receiving systemic immunosuppression, thus preventing extrapolation of the data to patients with ALD or immunosuppressed liver transplant recipients. Hence, significant information about liver safety profiles is mainly unreported, except that aberrant liver biochemistry was recorded in just one of 12,021 patients who received the Astra Zeneca vaccine [72]. The paper by Mahmud et al. was the first one to our knowledge that classified the patients based on their etiology (31.3% HCV–related liver disease, 33.0% alcohol-related liver disease, and 31.6% nonalcoholic fatty liver disease (NAFLD). In addition, the authors conducted an interesting retrospective cohort study to understand the impact of the Pfizer-BioNTech and Moderna mRNA vaccines in cirrhotic patients compared to a control group of unvaccinated patients without liver injury [72]. There were 43,122 patients included. The authors reported that one shot of an mRNA vaccine was related to a 64.8% reduction in SARS-CoV-2 infections and 100% protection against hospitalization or death due to COVID-19 by 28 days following the initial dose. They also estimated that patients with decompensated cirrhosis had a lower rate of SARS-CoV-2 infections after the first dose (50.3%) than those with compensated cirrhosis (66.8%) [73].

Going further down the road, even fewer studies reported safety data from the vaccine in AIH patients and its association with the flare of the disease. For the time being, the available information came mainly from published case reports. For example, Torrente et al. reported a case of a 46-year-old female with an AIH controlled without medications who experienced a flare of the disease after the COVID-19 vaccination [74].

In addition, Cao et al. published a case report of vaccine-induced AIH exacerbation in a 57-year-old Asian female without previous medical history. However, they rejected the hypothesis of AIH onset following the COVID-19 vaccine because of the histological grade of the fibrosis (stage 2 in this case scenario), which could not be possible for such short notice [75].

Even though the literature data is scarce regarding liver transplant patients, a study by Callaghan et al. showed that the vaccination course does not decrease the number of COVID-19 patients [76]. Their research was conducted from September 2020 to August 2021. The authors included 577 liver recipients; 370 were unvaccinated, 33 received only 1 dose, and 174 patients had a full vaccination course. Nevertheless, it is crucial to underline that those who received two vaccine doses had a 20% higher probability of survival in case of contact with SARS-CoV-2 infection [76].

According to the American Association for the Study of Liver Disease (AASLD) and the European Association for the Study of the Liver (EASL), there should be a prompt SARS-CoV-2 vaccination in patients with advanced liver injury [77]. They also recommend against live virus vaccines in patients on high-dose corticosteroids or immunosuppression therapy [77,78].

However, the immediate and long-term protective response through immunization may be insufficient due to these patients’ impaired immune responses. Therefore, AASLD also recommends a booster dose of an mRNA vaccination for all immunocompromised people who have previously received two doses of an mRNA vaccine due to their decreased response rate and greater risk of breakthrough infections [78]. According to the AASLD expert panel consensus statement, the booster shot of the mRNA vaccine should be administered at least 28 days after the last dose and should be a homologous mRNA vaccine whenever possible. In areas where the homologous shot is unavailable, the alternative (heterologous) mRNA vaccine could be used if necessary [78]. In addition, in cases of a limited supply of COVID-19 vaccine, the patients with higher Model for Endstage Liver Disease (MELD) or Child–Turcotte–Pugh scores should prioritize vaccination [78].

The growing impact of SARS-CoV-2 immunization has aroused concerns about vaccine-induced side effects. Besides systemic adverse effects (AE) such as headache, fever, fatigue, and those reported from the clinical trials (Table 2), physicians should consider reactivation of occult autoimmune diseases as a vaccine side effect [79,80].

Three case reports in the literature described AIH development several days after COVID-19 immunization. Nonetheless, the exact mechanism has not been elucidated [81,82].

In the past, this association between vaccination and autoimmune disease has been linked to using different adjuvants such as aluminum hydroxide, Toll-like receptor agonists, or lipid emulsions in developing subunit and inactivated vaccines [83,84]. However, no causal relationship has been established, even for adjuvanted vaccinations, including aluminum hydroxide or aluminum phosphate [85].

Cross-reactivity with host cells is thought to be the main villain due to the disrupted self-tolerance and promoted autoimmune reactions because of the SARS-CoV-2 infection. Consistent with this report, COVID-19 mRNA vaccines may trigger a similar effect [4,86].

Regardless, prioritizing vaccination in this cohort remains critical, considering the high percentage of COVID-19-related mortality in patients with decompensated cirrhosis. Currently, recommendations for vaccination administration in disease subpopulations are inconsistent and geographically variable. A detailed study of immunological reactions must be performed to standardize vaccination guidelines eventually. As we enter a new phase of SARS-CoV-2 immunization, it is crucial to investigate the impact of recent vaccinations on patients with various liver diseases, where information is lacking. Still, the clinical effects of the viral infection are severe. Large epidemiological studies, well supported by pharmacovigilance trials, would be of paramount significance for evaluating vaccine safety profile, its AE rates, and its association with re-/activation of an autoimmune disease. Better engagement between scientists and physicians is needed to establish the best vaccine choice for different liver disease populations.

## 4. Pediatric Point of View on COVID-19 Vaccines in Children with Gut and Liver Autoimmune Diseases

Gastrointestinal involvement in children associated with SARS-CoV-2 is multifaceted. Acute infection is related to various GIT symptoms such as nausea, vomiting, diarrhea, and abdominal pain [87]. According to a recent systematic review of children with COVID-19, the pooled prevalence of diarrhea is 12.4%, followed by vomiting at 10.3%, and vomiting at 5.4% [87]. In some cases, SARS-CoV-2 RNA could be detected in feces and persist even after nasopharyngeal smears are negative for up to 12 days [88]. A more prominent abdominal symptom is seen in MIS (PIMS) children [89]. In a multicenter study, Lo Vecchio et al. reported a 9.5% incidence of severe GI symptoms in Italian children with SARS-CoV2 infection. They included acute abdomen, appendicitis, intussusception, pancreatitis, abdominal fluid collection, and diffuse adenomesenteritis, frequently associated with MIS-C [89]. Acute COVID-19 infection and MIS-C could lead to a different degree of liver injury—almost always reversible and mild [90].

Children with advanced liver diseases (cirrhosis, portal hypertonia) and/or children on immunosuppressive therapy have an increased risk for severe COVID-19. Pediatric IBD (PIBD) with COVID-19 infection appears to be less severe than in adults. Children with IBD, with or without biological and/or immunosuppressive treatment, do not demonstrate a higher risk during SARS-CoV-2 infection than the general population. However, many studies emphasize the role of uninterrupted PIBD therapy since its discontinuation could cause a disease flare during COVID-19 infection [91]. On treatment with infliximab (anti-TNF therapy) as monotherapy or combination therapy, children with IBD probably will need a booster dose to provide complete protection [92]. Spencer et al. report stable seroconversion in children with PIBD after SARS-CoV-2 infection and vaccination [93].

Considering IBD children who are more prone to infections, strict routine vaccines—flu and pneumococcal vaccines—are recommended. This should also be applied to COVID-19 vaccines, although IBD children have no tendency for a severe course of disease than the general pediatrician population. However, there is an urgent need for official, internationally accepted recommendations for SARS-CoV-2 vaccination in children with IBD and other autoimmune gastrointestinal diseases. According to the advice from an international consensus meeting on SARS-CoV-2 vaccination for patients with IBD, the immunization of children with IBD will be the same as in the general pediatric population after the official SARS-CoV-2 vaccination authorization [94]. It is expected that COVID-19 vaccines would elicit an adequate immune response since immunosuppressive treatment does not reduce vaccine immunogenicity in IBD children [95].

The Crohn’s and Colitis Canada organization officially recommends that children five years of age or older with IBD receive their second dose of COVID-19 vaccination approximately 4 weeks after their first dose and their third complete COVID-19 vaccination 4–8 weeks after their second vaccine dose [96]. However, the organization also comments on the possible fourth dose in children without establishing official recommendations and specific conditions. Therefore, there are still no recommendations for the fourth dose in children aged 5–11. Additionally, after initiating mass child vaccination worldwide, no recommendations were updated for 2022.

According to a joint ESPHAGAN (European Society for Pediatric Gastroenterology, Hepatology and Nutrition) and SPLT (Society of Pediatric Liver Transplantation) position paper, children and adolescents with cirrhosis (compensated and decompensated), chronic liver disease (CLD) (including nonalcoholic fatty liver disease), and end-stage liver disease awaiting transplantation, as well as liver transplantation (LT) recipients on immunosuppressive medications, should be prioritized for early vaccine access because of their risk for poorer outcomes from SARS-CoV-2 infection [97]. Furthermore, children listed for LT can receive a two-dose-schedule vaccination. Even LT is planned between two doses. Additionally, it is not recommended to reduce immunosuppression to elicit an immune response after vaccination, nor serological testing before and after the COVID-19 vaccine. The position paper recommends SARS-CoV-2 vaccination for children 12–17 years old. After evaluating the vaccine’s safety, the same recommendation should be applied to younger children with different chronic liver diseases based on the expected benefits and confirmed safety [97]. However, ESPGHAN recommendations are not updated after the official approval from FDA and EMA for mRNA COVID vaccines in children from 5–11 years [91].

To sum up, The Crohn’s and Colitis Canada organization alone provided specific recommendations for COVID-19 vaccination in children with IBD. However, there is a lack of official guidance from established international societies and organizations. In addition, we need details on vaccination dosage and regimens (including third and following doses of reimmunization) and specific situations, such as the administration of systemic corticosteroids and immunosuppressants, including biologics.

Dailey et al. provided valuable information on the efficacy of COVID-19 vaccination on children and young adults with IBD (aged 2–26 years) [28]. As expected, and similar to the results from adults with IBD, antispike IgG antibodies were significantly lower in patients with IBD after natural infection and there was a 15-fold increased antibody response following COVID-19 vaccination. Nevertheless, after vaccination, all patients developed virus-neutralizing antibodies [28].

We also must emphasize that parents and legal guardians of children with autoimmune gut and liver diseases have concerns and fears about the SARS-CoV-2 and vaccination against it. These concerns lead to hesitation on how to proceed with immunosuppressive therapy. If we extrapolate the data for adults, it is mandatory to maintain treatment during vaccination to avoid relapses. Therefore, physicians must support them and provide the proper information to explain all the benefits and safety profiles of COVID-19 vaccines. However, all specialists need specific official recommendations and statements from the official organizations and societies for children with this condition, which should also be regularly updated considering the novel accumulated data.

## 5. Conclusions

The discussion about COVID-19 vaccines and GI patients with altered immune status resulted in an evidence gap for the long-term safety and efficacy of COVID-19 vaccines in patients with autoimmune diseases. In addition, some of the theoretical concerns are related to the complex pathogenesis of autoimmune diseases, including liver and gut conditions, immunosuppressive therapy, and the cumulative risk of flare. Nevertheless, data on COVID-19 vaccination for adult and pediatric patients with gut and liver autoimmune diseases showed good efficacy of mRNA vaccines. Moreover, their immunogenicity and safety profile are acceptable, with adverse effects resembling those in the general population. Still, there is a lack of official statements and recommendations, and more clinical trials are needed to confirm these data and support official recommendations.

## Figures and Tables

**Table 1 vaccines-10-02075-t001:** Summarized COVID-19 vaccine studies on safety profile, efficacy, and adverse effects rate in patients with inflammatory bowel disease.

Type of Vaccine	Type of Study	Subjects	Data on Efficacy (% Protection, Other)	Data on Safety (Main Side Effects)	Reference
mRNA	Prospective study design	All patients included n = 246 (67% Crohn’s disease, 33% ulcerative colitis)	N/A	After the first dose (injection site reactions in 38%; fatigue/malaise 23%, headaches 14%, fever/chills 5%); After the second dose (injection site reaction 56%; fatigue, malaise 45%, headaches 34%, fever/chills 29%)	Botwin et al. [19]
mRNA, adenoviral	A prospective, observational cohort study	All patients included n = 3316 with IBD (n = 1908, Pfizer/BioNTech; n = 1272 Moderna, n = 161, Janssen)	N/A	No severe systemic reactions require emergency room visits. After the first dose: adverse reaction injection site (66%); fever (6%), fatigue (46%), headaches (32%), muscle aches (20%); After the second dose: adverse reaction injection site (65%); fever (25%), fatigue (46%), headaches (32%), muscle aches (12%); Low flare rate (2%)	Weaver et al. [20]
mRNA	Self-reported study	84 IBD patients (23-with Crohn’s disease, 25 with ulcerative colitis) on anti-TNF therapy	Biologic therapy associated with lower anti-RBD antibodies	N/A	Wong et ICARUS-IBD Working Group [21]
mRNA	Multicenter, UK prospective, case-control study	352 IBD patients on immunosuppressive therapy (thiopurine, infliximab, ustekinumab, vedolizumab, tofacitinib) and 72 healthy controls	No significant differences in anti-SARS-CoV-2 S1 RBD antibody concentrations between the healthy control group and patients treated with thiopurine, ustekinumab, or vedolizumab, lower anti-SARS-CoV-2 S1 RBD antibody concentrations independently associated with infliximab, tofacitinib, and thiopurine, but not with ustekinumab or vedolizumab (0.84 [0.54–1.30]; *p* = 0.43)	N/A	Alexander et al. [22]
mRNA	Multicenter Israeli population-based cohort study	12,109 IBD patients, 4946 non-IBD controls, 707 unvaccinated IBD patients	99.7% protection; patients on TNF inhibitors and/or corticosteroids did not have a higher incidence of infection; risk of exacerbation was 29% in vaccinated vs. 26% in unvaccinated IBD (*p* = 0.3)	N/A	Lev-Tzion et al. [23]
mRNA and adenoviral	Prospective, CLARITY IBD multicenter cohort study	1293 vaccinated IBD patients	anti-SARS-CoV-2 antibody concentrations reduced in patients treated with infliximab than vedolizumab	N/A	Kennedy et al. [24]
mRNA	Retrospective	7321 vaccinated IBD, 7376 unvaccinated IBD patients	Full vaccination associated with 69% reduced risk for COVID-19 and 80.4% effectiveness	N/A	Khan and Mahmud [25]
mRNA	Retrospective	5562 vaccinated IBD, 859,017 vaccinated non-IBD patients	N/A	2.2% adverse events in IBD patients on biologics/immunomodulatory therapy vs. 1.67 without such treatment; special adverse events	Hadi et al. [26]
mRNA	Retrospective cohort	12,231 vaccinated IBD, 36,254 vaccinated non-IBD patients	0.19% breakthrough infections after the second dose (7 days) and 0.14% (14 days)	N/A	Ben-Tov et al. [27]
mRNA, adenovirus vector	Prospective	33 vaccinated IBD—children and young adults	15 times higher levels of IgG antibodies compared to natural infection, all participants developed neutralizing antibodies	For mRNA vaccine—sore arm, chills, fever, etc.; vector vaccine—the same; no one has contracted COVID-19 2–6 months following vaccination	Dailey et al. [28]
mRNA	Prospective single-center	317 vaccinated IBD patients	Detectable antibodies in 300/317 IBD patients; 85% in patients on corticosteroids	N/A	Kappelman et al. [29]
mRNA	Prospective, multicenter	84 patients with immune-mediated disease, 8 vaccinated IBD patients	90.5% of all patients with immune-mediated diseases develop IgG antibodies to SARS-CoV-2	Less frequent mild adverse effects (injection site pain, headache, chills, arthralgia)	Simon et al. [30]
mRNA	Prospective, multicenter	133 patients with chronic inflammatory disease, 42 vaccinated IBD patients	N/A	Incidence rate of overall adverse events—0.55; local—0.64; mainly fatigue, headache, myalgia, fever and chills; severe adverse reactions incidence rate 0.02, requiring hospitalization—0.00 and IBD flares—0.01	James et al. [31]
mRNA, adenovirus vector	Prospective, multicenter	353 vaccinated IBD patients	Higher quantitative log10 antispike IgG after mRNA vs. adenovirus	N/A	Pozdnyakova et al. [32]
mRNA	International web-based survey	3272 IBD patients	N/A	72.4% local symptoms, 51.4% systemic symptoms	Ellul et al. [33]
mRNA	Cohort study	122 IBD patients and 60 controls, on immunomodulating therapy	97% of IBD patients developed antibodies, lower in patients than in controls, higher after Moderna vs. Pfizer; lower when on immunosuppressive therapy;	OR = 0.97 significant side effects associations after full vaccination	Caldera et al. [34]
mRNA, adenovirus vector	Prospective single-tertiary study	602 IBD patients on immunosuppressive therapy	Lower Ig concentrations in patients on treatment; 97.8% seropositivity in IBD patients	N/A	Cerna et al. [35]
mRNA, adenovirus vector	Retrospective observational	72 IBD patients;	100% antibody response in patients group; reduced antibody levels in IBD vs. controls, no differences between vaccines; all IBD patients developed an immune response	Local and systemic mild reactions	Classen et al. [36]
mRNA	Prospective controlled	185 IBD patients, 73 healthy controls	100% response following vaccination, lower in older and on anti-TNF therapy	Local pain, headache	Edelman-Klapper et al. [37]
mRNA, adenovirus vector	Cohort/ real-life survey: telephone questionnaire	239 IBD patients on biologics	N/A	High acceptance rate and mild and transitory adverse reaction	Garrido et al. [38]
mRNA	Prospective study	19 IBD patients on biologics	95% immune response rate	N/A	Levine et al. [39]
mRNA	Prospective study	19 patients on biologics	21.13-fold increase of total IgG antibodies after 1st dose, and 90-fold after second dose; % virus neutralizing antibodies was lower in IBD patients	N/A	Rodriguez-Martino et al. [40]
mRNA, adenovirus vector	Prospective study	126 IBD patients on biologics	74.5–81.2% immune response in patients on anti-TNF vs. 92.8–100% on vedolizumab and ustekinumab, resp.; fewer virus-neutralizing antibody titers in patients treated with anti-TNF	N/A	Shehab et al. [41]

**Table 2 vaccines-10-02075-t002:** Summarized COVID-19 vaccine studies on safety profile, efficacy, and adverse effects rate in patients with autoimmune liver disease.

Type of Vaccine	Type of Study	Subjects	Data on Efficacy (% Protection, Other)	Data on Safety (Main Side Effects)	Reference
mRNA	Placebo-controlled, observer-blinded, pivotal efficacy trial (randomized 1:1 vaccine vs. placebo)	All patients included n = 43,548 Patients with liver disease n = 217 (0.6%)	95% efficacy (9 vaccinated vs. 169 controls with COVID-19)10 cases of severe COVID-19 infection vs. 9 in the placebo group	Systemic AEs: 1. Fatigue (34–51%) 2. Headache (25–39%) 3. Fever (11%); Injection site reactions: 4. Pain (71–83%) 5. Redness and swelling (<7%) Serious AE <4%. Flares: NR	Polack et al. [68]
mRNA	Phase 3 randomized, observer-blinded, placebo-controlled trial was conducted at 99 centers across the United States (randomized 1:1 vaccine vs. placebo)	All patients included n= 30420 Patients with liver disease n = 196 (0.6%)	94.1% efficacy (11 vaccinated vs. 185 controls with COVID-19)None of the patients were with COVID-19 infection vs. 30 cases of severe COVID-19 in the placebo groupFlares: NR	Systemic AEs: 1. Fatigue (34–38%) 2. Headache (24–35%) 3. Fever (<1%); Injection site reactions: 4. Pain (86%) 5. Redness and swelling (<6%) Serious AE: Low rate after the first dose and increased to around 16% after the second dose	Baden et al. [69]
mRNA, adenovirus vector	Observational study	103 patients with autoimmune hepatitis, 64 with primary sclerosing cholangitis, 61 with primary biliary cholangitis, 95 healthy controls	Anti-SARS-CoV-2 antibodies were relatively lower in patients with autoimmune hepatitis vs. healthy control and comparably low in patients on immunosuppression; a spike-specific T cell responses were undetectable in 45% of patients despite a positive serology in hepatitis patients and 87% in primary sclerosing cholangitis and primary biliary cholangitis	N/A	Duengelhoef et al. [70]
Adenoviral vector	Blinded, randomized, controlled trials conducted across the U.K., Brazil, and South Africa.	All patients included n = 11,636 Patients with liver disease—NR	70.4% efficacy (30 vaccine recipients vs. 101 placebo-group)	84 serious AEs in the vaccine group	Voysey et al. [71]

## Data Availability

Not applicable.

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
