# Peer review of "COVID-19 Vaccines for Adults and Children with Autoimmune Gut or Liver Disease"

_vaccines, 2022, doi:10.3390/vaccines10122075_

Round 1
Reviewer 1 Report
Comments and Suggestions for Authors
There have been hesitancies concerning the safety and efficacy of Covid-19 vaccines for patients with autoimmune gut or liver diseases. This mini-review manuscript is addressing these important issues.
Despite their comprehensive review discussing this topic, there are some minor issues that should be solved or addressed before being appropriate for publication.
The minor issues:
Abstract: The abstract is well written, addressing the topic, and describing the issues with a good conclusion. In line 26, the abbreviation of IBS should be described, in the first place of mentioning. I see that is described in line 27 (irritable bowel syndrome).
1. Introduction: This manuscript has focused the gene vaccines, ignoring the protein vaccines which have been used widely in several countries. Is there any reason not to mention them?
The authors should describe different types of COVID-19 vaccines; live-attenuated vaccines, protein-based vaccines, and gene vaccines (mRNA, Vector-based, and VLPs). There are some studies reviewing the safety and efficacy application of VLPs for Covid19 vaccine: (https://doi.org/10.3390/vaccines9121409), and also for COVID-19 vaccine classification (https://doi.org/10.3390/vaccines10101655).
2. Autoimmune gut diseases and COVID-19 vaccines
Table-1 is not well designed. The column showing the “Authors” should be the last column. Please transfer it to the right side.
In the last column (Data on safety), the way of numbering the items is very confusing, and should be “aligned to the left”.
Furthermore, there are 2 sections in this table, representing “Gut diseases” and “Liver diseases”. Please either split these 2 sections or make them bold, with clear splitting lines. Otherwise, these 2 sections are not well observable at the first glance.
There are several typos and mistypes in the table. Please correct them. Some examples are:
In first row: fatigue/malaise 23#,
In the last row: Voysey et al. [36] is not in the correct column (Authors).
5. Conclusions
Please replace the term “sensitive issues” with an appropriate term.
In line 335, this sentence is too long and confusing and should be either shortened or split into 2 sentences.
“Nevertheless, data on COVID-19 vaccination …”
Author Response
Dear reviewer,
Thank you for your time to revise our Manuscript: COVID-19 vaccines for adults and children with autoimmune gut or liver disease
We have incorporated the suggestions made by you. Please see below, in blue, for responses to your comments. All page numbers refer to the revised manuscript file with highlighted changes.
Comments and Suggestions for Authors
There have been hesitancies concerning the safety and efficacy of Covid-19 vaccines for patients with autoimmune gut or liver diseases. This mini-review manuscript is addressing these important issues.
- Thank you very much for the overall evaluation of paper as good.
Despite their comprehensive review discussing this topic, there are some minor issues that should be solved or addressed before being appropriate for publication.
The minor issues:
Abstract: The abstract is well written, addressing the topic, and describing the issues with a good conclusion. In line 26, the abbreviation of IBS should be described, in the first place of mentioning. I see that is described in line 27 (irritable bowel syndrome).
- Thank you very much for the valuable comment, we corrected the issue. We meant IBD, not IBS.
- Introduction: This manuscript has focused the gene vaccines, ignoring the protein vaccines which have been used widely in several countries. Is there any reason not to mention them?
The authors should describe different types of COVID-19 vaccines; live-attenuated vaccines, protein-based vaccines, and gene vaccines (mRNA, Vector-based, and VLPs). There are some studies reviewing the safety and efficacy application of VLPs for Covid19 vaccine: (https://doi.org/10.3390/vaccines9121409), and also for COVID-19 vaccine classification (https://doi.org/10.3390/vaccines10101655).
- Thank you very much on emphasizing this. We focused mainly on the mRNA vaccines due to the data on safety and effectiveness of these vaccines in the GE patients. However, we find your comment for valuable and we will add information and data on the other vaccines types as well, including description of their mechanisms in providing immune protection against COVID-19.
- Autoimmune gut diseases and COVID-19 vaccines
Table-1 is not well designed. The column showing the “Authors” should be the last column. Please transfer it to the right side.
In the last column (Data on safety), the way of numbering the items is very confusing, and should be “aligned to the left”.
Furthermore, there are 2 sections in this table, representing “Gut diseases” and “Liver diseases”. Please either split these 2 sections or make them bold, with clear splitting lines. Otherwise, these 2 sections are not well observable at the first glance.
There are several typos and mistypes in the table. Please correct them. Some examples are:
In first row: fatigue/malaise 23#,
In the last row: Voysey et al. [36] is not in the correct column (Authors).
- Thank you very much for all the valuable suggestions and recommendations to improve the quality of table 1. We revised the columns, corrected the mistakes. We believe that the table has been improved significantly regarding clarity, comprehension and informativeness.
- Conclusions
Please replace the term “sensitive issues” with an appropriate term.
- Thank you for the valuable comment, we changed it with the neutral “discussion about”.
In line 335, this sentence is too long and confusing and should be either shortened or split into 2 sentences. “Nevertheless, data on COVID-19 vaccination …”
- We agree that the sentence is unclear and difficult to understand. We divided the sentence into a few plain sentences.
Reviewer 2 Report
It seems to me that it would be advisable at the beginning of the article and then in the discussion to reflect the statistics of publications in relation to the topic under discussion - as publications of 2020, 2021 and 2022 reflect articles on vaccination of the category of patients under discussion, incl. in comparison with patients with other autoimmune diseases (rheumatic diseases, diabetes mellitus, etc.). In this section, it would be appropriate to discuss in more detail the mechanisms of the immune response to virus SARS-CoV-2 and to vaccination against it in patients with various immunoinflammatory diseases.Author Response
Dear reviewer,
Thank you for your time to revise our Manuscript: COVID-19 vaccines for adults and children with autoimmune gut or liver disease
We have incorporated the suggestions made by you. Please see below, in blue, for responses to your comments. All page numbers refer to the revised manuscript file with highlighted changes.
It seems to me that it would be advisable at the beginning of the article and then in the discussion to reflect the statistics of publications in relation to the topic under discussion - as publications of 2020, 2021 and 2022 reflect articles on vaccination of the category of patients under discussion, incl. in comparison with patients with other autoimmune diseases (rheumatic diseases, diabetes mellitus, etc.). In this section, it would be appropriate to discuss in more detail the mechanisms of the immune response to virus SARS-CoV-2 and to vaccination against it in patients with various immunoinflammatory diseases.
- We agree with the referee that it would be good to discuss also the data on vaccination of the category of patients, incl. in comparison with patients with other autoimmune diseases (rheumatic diseases, diabetes mellitus, etc.), as well as the mechanisms of the immune response to virus SARS-CoV-2 and to vaccination against it in patients with various immunoinflammatory disease. We added additional information and more studies to cover 2020-2022 publications.
- We provided data from COVAD study (COVID-19 vaccination in patients with autoimmune diseases) recent publications, and more articles on the topic.
Reviewer 3 Report
In this review, the authors discussed COVID19 vaccine and autoimmune diseases of gut and liver diseases. The authors presented data of the vaccine in autoimmune gut disease. But the point was discussed in several reviews before, and there are several missing points
Major points
a) For COVID19 in IBD: there are missing data: PMID: 34954338,
b)The effect of COVID19 vaccine in IBD in immunosuppressed patients. PMID: 36088954
c) In this paper PMID: 35987619, table #2 there is a comparsion between different COVID19 vaccines
d) Also another manuscript described this message
https://www.ncbi.nlm.nih.gov/pmc/articles/PMC9143939/
e) Also this article described a lot about of COVID19 and liver disease
PMID: 36229799
Author Response
Dear reviewer,
Thank you for your time to revise our Manuscript: COVID-19 vaccines for adults and children with autoimmune gut or liver disease
We have incorporated the suggestions made by you. Please see below, in blue, for responses to your comments. All page numbers refer to the revised manuscript file with highlighted changes.
In this review, the authors discussed COVID19 vaccine and autoimmune diseases of gut and liver diseases. The authors presented data of the vaccine in autoimmune gut disease. But the point was discussed in several reviews before, and there are several missing points
- Thank you for the overall evaluation of our paper as good. We acknowledge all the critical comments to improve the quality of our manuscript.
Major points
- a) For COVID19 in IBD: there are missing data: PMID: 34954338,
- Thank you very much for the valuable information. We included data from this study in the manuscript.
b)The effect of COVID19 vaccine in IBD in immunosuppressed patients. PMID: 36088954
- This group of IBD patients is critical to be included in the evaluation of the vaccination in patients with AID, especially those who are compromised.
- c) In this paper PMID: 35987619, table #2 there is a comparsion between different COVID19 vaccines
- Thank you very much for the valuable suggestion and recommendation, it seems that we`ve missed this paper due to its publishing in the late summer this year. We included data from it regarding different types of COVID-19 vaccines in IBD patients.
- Motivated by your critical comments, we performed new search for data regarding IBD COVID-19 vaccination, and add it to the paper. We accept that many systematic reviews and metaanalyses were published on the topic of IBD and liver disease. However, still, our paper is focused on both groups, including pediatric patients, therefore, this is one of the strengst of our narrative review.
- d) Also another manuscript described this message
https://www.ncbi.nlm.nih.gov/pmc/articles/PMC9143939/
- e) Also this article described a lot about of COVID19 and liver disease
PMID: 36229799
- Thank you very much for the recommendations, we included all the suggested papers, as well as the other that we found in the indexed journals.
Round 2
Reviewer 3 Report
The manuscript is improved and the authors addressed some of my suggestions.